# Inductive Representation Learning of Temporal Pattern Subgraphs in Temporal Networks

## Abstract

Subgraph structure learning on graph neural networks (GNN) has attracted considerable attention recently because of its capacity to encode high-level graph structural features. Temporal network representation learning has been used in many real-world dynamic systems that usually evolve according to some temporal patterns, such as the triadic closure laws in social networks. Inductive representation learning of temporal networks should be able to capture these temporal patterns and further apply them to nodes which were not discovered during the training. Previous work neglected to extract these patterns, or could not be applied to downstream tasks because of the high time complexity of matching. In this paper, we design the strategy for capturing two types of prevalent temporal patterns. We propose the TPSN framework for inductive temproal pattern learning, in which we perform adjacent edge reconstruction on the extracted subgraphs, thereby improving the learning efficiency of temporal triadic closure laws and reducing the possibility of oversmoothing. Furthermore, we use multi-subgraph contrastive learning to achieve higher accuracy with fewer negative samples. Our proposed method outperforms baselines in all three downstream tasks and maintains acceptable time complexity. Ablation experiments also validate the effectiveness of our proposed model and module.

## 1 Introduction

Graph neural networks (GNNs)Scarselli et al. (2008) have achieved significant success because of their unique message-passing capability, which aims to continuously aggregate information from neighbors. To date, GNNs have shown excellent performance in a wide range of applications, such as node classificationMonti et al. (2017)Kipf & Welling (2016a) and link predictionZhang & Chen (2018). Among them, temporal networksHolme (2015)Holme & Saramäki (2019) are developing rapidly. Temporal networks, as opposed to static networks, take into account the effect of temporal factors on nodes and links, and hence can be used to represent more complicated relationships, for instance, in social networksHe & Chen (2015) and citation networks.

According to the law of triadic closure, two nodes in a social network with common neighbors tend to get to communicate strongly. It is known that the interaction behavior of a node is influenced by its local unclosed triads, recent interactions, and the evolutionary patterns of neighboring nodes with frequent interactions. Most previous research relies on static GNNs to process data from temporal sequences and subsequently create GNN models. It is well known that GNN model performance is closely related to the quality of the given graphs. GNNs simply seek to memorize current structures and do poorly while learning an implicit representation. Due to the complexity and denseness of real-world social datasets and that the given graphs are often contaminated by noise. This adversely affects both GNN recognition performance and the extraction of interaction evolutionary patterns. There is also a problem of combinational explosion.

Therefore, many temporal GNNs attempt to model dense temporal graphs with frequent interactionsGoyal et al. (2018)Nguyen et al. (2018)Pareja et al. (2020)Xu et al. (2020)Wang et al. (2021). To recognize temporal pattern subgraphs, most work uses the temporal motif mining approachShao et al. (2012)Du et al. (2009)Shao et al. (2013), which searches the complete network for pre-defined temporal motifs and counts their frequency. The method then extracts higher-order features and uses them for downstream processing. For the realistic downstream tasks, this method is often in-

sufficient. Although various algorithms have been presented to increase the efficiency of mining temporal motifsSun et al. (2019)Liu et al. (2019), obtaining temporal motifs with three or more nodes or obtaining several temporal motifs at the same time is still time-consuming. Moreover, unlike static networks, temporal networks evolve with time. That means that motif-mining should be re-performed when the temporal network changes and then causes high time complexity. Instead, static motifs are to frequently used for temporal networks. For example, Liu et al.Liu et al. (2021) use Parameterized Graphlet(PGD) to obtain motifs for each snapshot. This effectively reduces the time complexity, but the time factors and the impact of interaction sequences on the evolution of social patterns are both ignored.

Our contributions in the paper can be summarized as follows: (1) We design a strategy for capturing two types of temporal pattern subgraphs which can represent the laws of node evolution. (2) We restructure the adjacency of extracted temporal pattern subgraphs based on the law of triadic closure to improve the efficiency of information propagation. This also reduces the complexity and avoids oversmoothing with the increase of layers. (3) We propose the multi-subgraph comparative learning method based on reconstructed temporal pattern subgraphs, which increases the prediction accuracy without increasing the number of negative samples required.

## 2 RELATED WORK

Relevant work mainly includes that on temporal network representation learning and temporal structured learning.

**Temporal Network Representation Learning**: Network Representation Learning, also known as graph embedding, is often used to transform large networks into lower-dimensional vectors. Recently, it has attracted significant attention. But most research focuses on static networks, and studies about temporal networks are relatively limited. CTDNEsNguyen et al. (2018) learns node embedding from a continuous-time dynamic network instead of a sequence of snapshots. JODIEKumar et al. (2019) uses two recurrent neural networks (RNNs) to learn trajectories of users and items, and updates the embedding when interaction occurs. Similar to JODIEKumar et al. (2019), DyRepTrivedi et al. (2019) also learns neighbor information. It captures topological evolution and activities between nodes to update the node representations. However, both JODIEKumar et al. (2019) and DyRepTrivedi et al. (2019) pay little attention to the frequency of interaction. As a result they may not effectively identify important nodes. VGRNNHajiramezanali et al. (2019) is a hierarchical variational model for temporal networks. EvolveGCNPareja et al. (2020) uses an RNN to evolve a graph convolutional network (GCN) parameters. TGATXu et al. (2020) utilizes a self-attention mechanism and presents a novel encoding method to learn graph embedding inductively. Learning temporal structure is therefore a significant part of temporal network representation learning.

**Temporal Structured Learning** : Structured Learning plays an important role in static network representation learning. However, temporal networks are often more complex and it is more difficult to capture temporal structure efficiently and then incorporate it into the learning process. Recently, several methods have been developed. Paranjape et al.Paranjape et al. (2017) propose the notion of a temporal network motif. They define it as the elementary unit of temporal networks. DynamicTriadZhou et al. (2018) is based on the changing configuration of the triad. It models the interconnectedness of three nodes to obtain the embedding at different times. It only uses information from the triads and does not fully learn the overall topology. MTSNLiu et al. (2021) uses the PGD technique to extract the motifs at each snapshot. It extracts structural features of each snapshot but ignores their temporal evolution. TM-MinerSun et al. (2019) introduces a labeling system to settle temporal graph isomorphism problem and further reduce the time complexity of motif mining. However, it is difficult to mine motifs with too many edges. All of the above methods capture part of the temporal structure, but do not focus on both the node features and the evolution of the network. As a result, they might not preserve information required for downstream processing task.

## 3 PROPOSED WORK

In this section, we elaborate on the proposed model TPSN, and the overall architecture is shown in Figure. 1. We first define the Temporal Pattern Subgraph and then propose two types of temporal

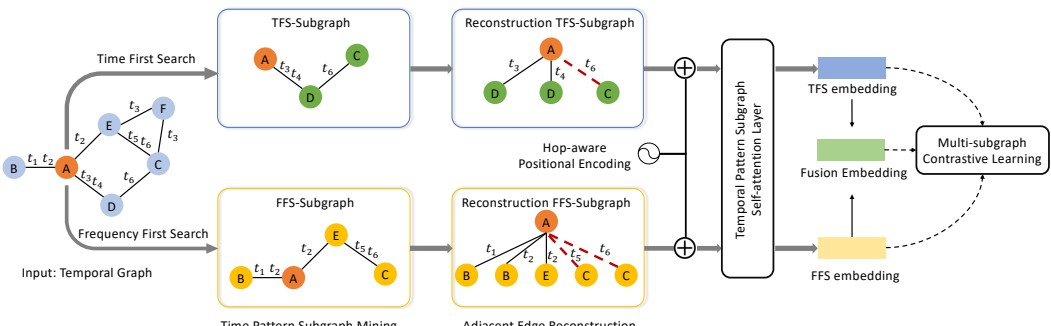

Figure 1: Framework of TPSN from the perspective of node A. TFS and FFS subgraphs are mined and reconstructed with hop-aware positional encoding, then encoded by a self-attention layer to produce TFS, FFS, and fusion embeddings, which are optimized through multi-subgraph contrastive learning.

pattern subgraph mining methods. Next, we perform an adjacent edge reconstruction of the temporal pattern subgraph. We design an aggregation strategy for information fusion with two different types of reconstructed time evolution subgraph. Finally, we use the multiple temporal subgraph representations of nodes for comparative learning.

### 3.1 TIME PATTERN SUBGRAPH MINING ALGORITHM

The mining algorithm of the temporal pattern subgraph aims to a) capture the real interaction evolutionary patterns of nodes, b) filter noisy edge interference, and c) improve the robustness of the model with maintaining low complexity. In the temporal graph, two nodes can interact multiple times at different instants in time, unlike in the static graph where each edge only appears once. Due to the huge number of time edges with unique time stamps, using the adjacency matrix will result in huge space consumption. Therefore, the triples are usually used to represent edges such as $(v_i, u_i, t)$ in which $v_i$ and $u_i$ are node indices ,$t$ is the time stamp of the edges. Adjacent relationships for the nodes are represented by adjacency lists. Consequently, given a dense temporal graph $G = (V, E, T)$ ,which can be represented as a sequence of links that change over time, i.e. $G = \{(v_i, u_i, t)|v_i, u_i \in V, t \in T\}$ where the set of neighbors of $v_i$ before the instant $t_i$ can be expressed as $AdjacencyList(v_i, t_i) = \{(u_j, t_j)|t_j < t_i, (v_i, u_j, t_j) \in G\}$. First, we use a temporal neighborhood finding algorithm to find $AdjacencyList(v_i, t_i)$, and recursively locate the k-hop neighbors of $vi$. The details of this algorithm are shown in Algorithm. 1 .

Then, we match the different types of subgraph to represent two temporal patterns within the K-hop neighborhood of nodes as follows:

#### 3.1.1 TFS-SUBGRAPH DETECTION

The behavior patterns of people in social networks are affected by their recent interactive evolution patterns. For example, the cascading of current notifications within an organization, the propagation of unexpected events. To extract this pattern, we use time-first search to find the subgraph in which interactions are closest to the current instant. This step is shown in Algorithm.2. We sort the edges of k-hop neighbors into a temporal order and add the top $N$ edges to the TFS-Subgraph, where $N$ is the edge size of the TFS-Subgraph.

#### 3.1.2 FFS-SUBGRAPH DETECTION

Human social behavior is also influenced by neighbors who maintain high-frequency interactions. The strong links between nodes cannot be ignored because of the absence of recent interaction or subside with the passage of time, such as fixed meeting matters with periodicity, or shopping interactions caused by regular discounts. Similar to TFS-Subgraph, for this evolutionary pattern, we use frequency-first search to find the subgraph in which nodes that most frequently interact with the same node.The details are shown in Algorithm.3.

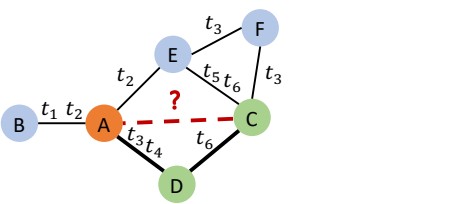 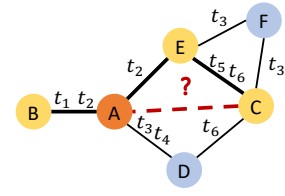

(a) TFS-Subgraph of node A with max 2-hop      (b) TFS-Subgraph of node A with max 2-hop

Figure 2: The TFS-Subgraph for node $A$ is shown in (a), and the FFS-Subgraph is shown in (b). From the law of triadic closure, node $A$ and node $C$ have a strong structural association. Our subgraph mining algorithm can capture the law of triadic closure through two different paths.

In terms of time complexity, the temporal pattern subgraph mining algorithms complete the matching of a specific subgraph pattern in $O(n^2)$, where $n$ is the average number of node links. Compared with the motifs mining algorithms with higher time complexity, in most situations, our temporal pattern subgraph mining algorithms are more suitable for real-time link change prediction for downstream tasks.

### 3.2 ADJACENT EDGE RECONSTRUCTION

As shown in Figure. 2a and Figure. 2b, we use temporal pattern subgraph mining algorithms in section 3.1 to find the TFS-Subgraph and FFS-Subgraph of node $A$. Node $A$ and node $C$ have a strong structural association according to the law of triadic closure in social networks. Our subgraph matching algorithm can capture this strong structural association through two different paths. But this is not obvious in the layer-by-layer aggregation of nodes. In order to improve the learning efficiency of the temporal triadic closure law and reduce the possibility of oversmoothing, we propose adjacent edge reconstruction for the temporal pattern subgraph.

#### 3.2.1 RECONSTRUCTING ADJACENT EDGES

First, we obtain the FFS-Subgraph and TFS-Subgraph for node $A$ using the temporal pattern subgraph mining method in Section 3.1. To facilitate the description, we define the aggregated path distance between two nodes as the number of edges traversed in the specified aggregated path. We also defined the FFS-Subgraph of node $A$ as $G_f$ and the TFS-Subgraph of node $A$ is $G_t$. Then for each edge $(v_i, u_i, t)$ in $G_f$, suppose the aggregated path distance between node $v_i$ and node $A$ is closer than the distance of node $u_i$ and node $A$. We replace node $v_i$ with node $A$ in the edge to form a new virtual edge $(A, u_i, t)$ directly connected to A. We also apply the same operation to $G_t$ for adjacent edge reconstruction.

#### 3.2.2 HOP-AWARE POSITIONAL ENCODING

Since the reconstructed edges are directly connected to node $A$, we use hop-aware positional encoding for the reconstructed edges to label the original aggregated path distance in the subgraph. The positional encoderVaswani et al. (2017) is as follows:

$$
\begin{cases}
PE(pos, 2i) = \sin\left(\frac{pos}{10000^{2i/d_{node}}}\right) \\
PE(pos, 2i+1) = \cos\left(\frac{pos}{10000^{2i/d_{node}}}\right)
\end{cases}
\tag{1}
$$

where $pos$ is the original aggregated path distance between node $A$ and its neighbor in the subgraph, and $d_{node}$ is the dimension of node embedding. We use $d_{node}$ as an edge feature, to augment to the representation of the node.

### 3.3 TEMPORAL PATTERN SUBGRAPH SELF-ATTENTION LAYER

The temporal subgraph embedding layer is used to generate the temporal embedding $h_i(t)$ of node $i$ at any time $t$ from the reconstruction subgraph. As shown in Figure. 3, for node A, the left side is

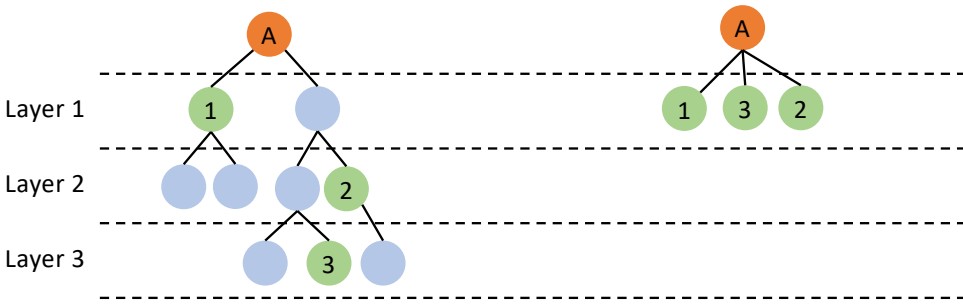

Figure 3: Node $A$ aggregation process of general method and ours. And the orange node $B$ is in node $A$'s FFS-Subgraph.

the general aggregation method to aggregate 3-hop neighbors, and the right side is our aggregation method with the reconstruction FFS-Subgraph and hop-aware positional encoding. Aggregating the neighbors in the range of 3-hops requires three GNN layers in the general aggregation method. We only use one self-attention layer to obtain node embeddings for temporal pattern subgraphs as follows:

$$\tilde{\mathbf{h}}_{i,FFS}^{(l)}(t) = f\left(\sum_{j \in G_f([0,t])} \left(\mathbf{h}_j^{(l-1)}(t) \,\|\, \mathbf{e}_{ij} \,\|\, (p(j) \,\|\, \phi(t - t_j))\right)\right) \quad (2)$$

$$\tilde{\mathbf{h}}_{i,TFS}^{(l)}(t) = f\left(\sum_{j \in G_t([0,t])} \left(\mathbf{h}_j^{(l-1)}(t) \,\|\, \mathbf{e}_{ij} \,\|\, (p(j) \,\|\, \phi(t - t_j))\right)\right) \quad (3)$$

where $\phi(*)$ represents the time encodingXu et al. (2020), $\mathbf{e}_{ij}$ represents the edge feature, and $p(*)$ is the hop-aware positional encoding. Function $f(*)$ is equivalent to conducting multi-head attentionXu et al. (2020). The node embeddings in TFS-Subgraph use the same way in another self-attention layer. As shown in Figure. 3, for reconstructed edges between node $A$ and the same neighbor node $B$, it is still possible to differentiate by time encoding and hop-aware positional encoding. In this way we can better distribute attention weights in the same layer. We fuse the node embeddings in two temporal pattern subgraphs by $MLP$. The fusion node embedding $\mathbf{Z}_i^{(l)}(t)$ is as follows:

$$\mathbf{Z}_i^{(l)}(t) = MLP(\mathbf{h}_{i,FFS}^{(l)}(t)) \| \mathbf{h}_{i,TFS}^{(l)}(t)) \quad (4)$$

### 3.4 MULTI-SUBGRAPH CONTRASTIVE LEARNING

For the multiple temporal pattern subgraph embedding, we use the triplet loss for multi-subgraph contrastive learning. We aim to make the nodes that are likely to be related in the future have a higher similarity among the temporal pattern subgraphs. If nodes are unlikely to be associated in the near future, they should have different evolutionary patterns. Multi-subgraph contrastive learning will make the learning of temporal patterns more effective and meaningful.

In contrastive learning, using more negative samples tends to have a better learning effect on the model. However, in temporal graph learning, the time complexity for neighbor finding is too large to use too many negative samples. Since multi-subgraph comparative learning can use two temporal pattern subgraph embeddings and one fusion embedding in each negative sample, our model achieves better results without adding more negative samples. First, we use the triplet loss function:

$$\ell(x_i^a, x_i^p, x_i^n) = \sum_i^N \Big[ \| f(x_i^a) - f(x_i^p) \|_2^2$$
$$- \| f(x_i^a) - f(x_i^n) \|_2^2 + \alpha \Big]_+ \tag{5}$$

where $\big[ \quad \big]_+$ indicates that if the value within $\big[ \quad \big]$ is greater than zero, the value is taken as a loss, and if it is less than zero, the loss is zero. The goal of the triplet loss is to make features with the same label as close as possible to each other in spatial location, while features with different labels are as far away as possible in spatial location. To prevent the features of the sample from aggregating within a very small space, for two positive sample and one negative case of the same class, the negative sample should be at least $\alpha$ farther away than the positive case, $\alpha$ is a hyperparameter. The final loss function is

$$L = -\log\Big( \sigma\Big( -\tilde{\mathbf{h}}_i^l(t_{ij})^\top \tilde{\mathbf{h}}_j^l(t_{ij}) \Big) \Big)$$
$$+ \ell(\mathbf{h}_{i,FFS}^{(l)}(t_{ij}), \mathbf{h}_{j,FFS}^{(l)}(t_{ij}), \mathbf{h}_{n,FFS}^{(l)}(t_{ij}))$$
$$+ \ell(\mathbf{h}_{i,TFS}^{(l)}(t_{ij}), \mathbf{h}_{j,TFS}^{(l)}(t_{ij}), \mathbf{h}_{n,TFS}^{(l)}(t_{ij})) \tag{6}$$

## 4 EXPERIMENTS

### 4.1 EXPERIMENTAL SETUP

#### 4.1.1 DATASETS.

We test the performance of our model on two benchmark:

*Reddit dataset*Kumar et al. (2019). This dataset includes user posting records on subreddits over a period of one month. The timestamp indicates the exact time of their posting. We select some of the more active users and their posting records from the original dataset to obtain a temporal graph with about 11,000 nodes and 700,000 edges. Through LIWCPennebaker et al. (2001), the features of these users are transformed into a 172-dimensional vector. The user label is then used to indicate whether they are banned or not.

*Wikipedia dataset*Kumar et al. (2019). This dataset records user editing of pages over the course of a month. The timestamp indicates the time when the user edited the page. We choose active users and frequently edited pages as nodes, and obtained a total of about 9000 nodes and 150,000 edges. As with the Reddit dataset, the features of these nodes were processed through LIWCPennebaker et al. (2001).The user labels indicate if users are temporarily banned from editing.

#### 4.1.2 EVALUATION TASK.

We use a) transductive link prediction, b) inductive link prediction and c) dynamic node classification for evaluation tasks.

*Transductive Link Prediction.*Transductive link prediction enables our model to learn links from all nodes during the training phase for a given time period and to predict the remaining links after that instant of time. It examines the model capacity for learning on seen nodes.

*Inductive Link Prediction.* Inductive link prediction requires our model to predict links between nodes that have not been seen in the training data. Compared with the transductive task, it focuses more on testing the performance of the model on implicit representation learning.We define two types of such linksWang et al. (2021): 1) "old-new" links, which are links between an observed node and an unobserved node and 2) "new-new" links, which are links between two unobserved nodes. They are used to evaluate the performance of different models on inductiveness.

*Dynamic Node Classification.* We use the node embedding after training to test node classification. For examples, in the Reddit dataset, dynamic labels indicating whether a user is banned from post-

Table 1: Performance in AUC for inductive link prediction

| Methods | Reddit | | Wikipedia | |
|---|---|---|---|---|
| | new - new | new - old | new - new | new - old |
| DynAERNN | 57.51 ± 2.54 | 58.79 ± 3.01 | 55.16 ± 1.15 | 57.97 ± 2.38 |
| JODIE | 72.49 ± 0.38 | 76.33 ± 0.03 | 70.78 ± 0.75 | 74.65 ± 0.06 |
| VGRNN | 61.93 ± 0.72 | 54.11 ± 0.74 | 60.64 ± 0.68 | 62.93 ± 0.69 |
| EvolveGCN | 63.31 ± 0.53 | 65.61 ± 0.37 | 58.01 ± 0.16 | 56.29 ± 2.17 |
| TGAT | 94.96 ± 0.88 | 97.25 ± 0.18 | 93.53 ± 0.84 | 95.47 ± 0.17 |
| TPSN(ours) | **98.03 ± 0.17** | **98.37 ± 0.12** | **96.73 ± 0.11** | **95.80 ± 0.23** |

Table 2: Performance in AUC for transductive link prediction

| Methods | Reddit | Wikipedia |
|---|---|---|
| DynAERNN | 83.37 ± 1.48 | 71.00 ± 1.10 |
| JODIE | 87.71 ± 0.02 | 88.43 ± 0.02 |
| VGRNN | 51.89 ± 0.92 | 71.20 ± 0.65 |
| EvolveGCN | 58.42 ± 0.52 | 60.48 ± 0.47 |
| TGAT | 96.65 ± 0.06 | 96.36 ± 0.05 |
| TPSN(ours) | **98.58 ± 0.10** | **96.75 ± 0.03** |

ing. In the Wikipedia dataset, dynamic labels indicate if users are temporarily banned from editing Xu et al. (2020).

### 4.1.3 BASELINES.

We select five state-of-the-art approaches for link prediction as our baselines. Depending on the input, these models can be divided into two categories: (1) Models for discrete representation, i.e. DynAERNNGoyal et al. (2020), VGRNNHajiramezanali et al. (2019) and EvolveGCNPareja et al. (2020); (2) Models for continues representation, i.e. JODIEKumar et al. (2019) and TGATXu et al. (2020). GAE, VGAEKipf & Welling (2016b), CDTNENguyen et al. (2018), GraphSAGEHamilton et al. (2017) and GAT are for node classification.

### 4.1.4 MODEL CONFIGURATION.

Hyperparameters control the two different types of temporal pattern subgraph construction, the sampling number of 1-hop neighbor $N$, the subgraph size $M$.Meanwhile. Using one TPSN layer and two attention heads gives the best validation performance. Use an early stopping strategy to select the best epoch to halt training and obtain the best model for the link prediction tasks. Finally, we use the node embeddings as features for dynamic node classifications with feedforward layers. The area under the ROC curve (AUC) is used to evaluate model performance.

## 4.2 RESULTS AND DISCUSSION

### 4.2.1 LINK PREDICTION PERFORMANCE.

We report the results of the transductive and inductive learning tasks in Table 2 and Table 1. The results demonstrate the state-of-the-art performances of our approach on both transductive and inductive learning tasks. In the inductive setting and especially with "new-new" links, our models significantly outperform all baselines on all datasets. Compared with TGAT which is the strongest baseline in AUC, we also uses the self-attention for node aggregation. In the experimental setup, TGAT uses two stacked attention aggregation layers, while the TPSN model uses only one attention aggregation layer and further reduces the time complexity.

### 4.2.2 DYNAMIC NODE CLASSIFICATION PERFORMANCE.

Since we use multi-subgraph contrastive learning, we can distinguish the dynamic label of a node more easily in different vector spaces through a) the node fusion embedding, b) TFS-Subgraph

Table 3: Performance in AUC for node classification

| Dataset | Reddit | Wikipedia |
|---|---|---|
| GAE | 58.39 ± 0.5 | 74.85 ± 0.6 |
| VGAE | 57.98 ± 0.6 | 73.67 ± 0.8 |
| CTDNE | 59.43 ± 0.6 | 75.89 ± 0.5 |
| GAT | 64.52 ± 0.5 | 82.34 ± 0.8 |
| GAT+T | 64.76 ± 0.6 | 82.95 ± 0.7 |
| GraphSAGE | 61.24 ± 0.6 | 82.42 ± 0.7 |
| GraphSAGE+T | 62.31 ± 0.7 | 82.87 ± 0.6 |
| TGAT | 65.56 ± 0.7 | 83.69 ± 0.7 |
| TPSN(ours) | **69.21 ± 0.7** | **86.51 ± 0.5** |

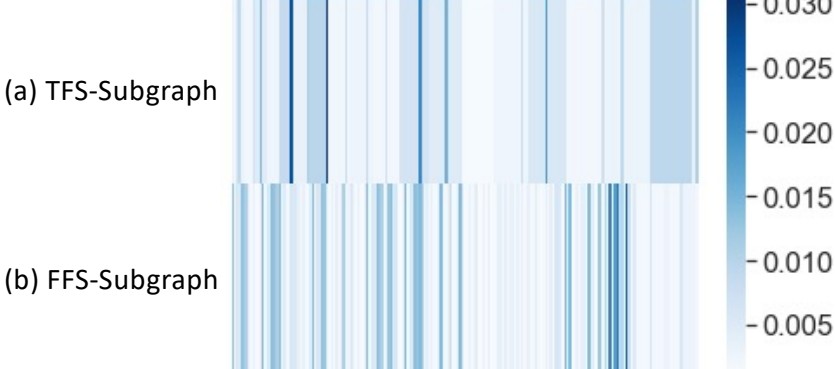

Figure 4: Attention weights in (a) TFS-Subgraph and (b) FFS-Subgraph.

embedding, c) FFS-Subgraph embedding. Moreover, we only sample one negative sample for each training sample pair. The results are shown in Table 3, our model exceeds all the baseline.

### 4.2.3 ATTENTION ANALYSIS.

We analyze how the attention weights of two types of temporal pattern subgraph contribute in the model. In Figure 4, we show the attention weights of the same node from Wikipedia for neighbors in different temporal pattern subgraphs. There are 196 edges in both the TFS-Subgraph and the FFS-Subgraph. The attention distribution of different temporal pattern subgraphs is different from a global perspective. Even for the same edges, the attention weights of the TFS-Subgraph are different from those of the FFS-Subgraph. This indicates that our model successfully distinguishes and leverages the two evolutionary patterns of the node.

### 4.3 ABLATION STUDY.

To validate the effectiveness of different parts in TPSN, we conducted an ablation study on the proposed TPSN model. We focus on ablation experiments for the temporal pattern subgraph as well as the effects of multi-subgraph contrastive learning, and we use AUC as our evaluation metrics.

### 4.3.1 VARIANTS OF TEMPORAL PATTERN SUBGRAPH.

To measure the importance of the temporal patterns gathered by our time pattern subgraph mining algorithm, we conduct ablation studies with the following temporal pattern variants, (a) only using TFS-Subgraph for TPSN, (b) only using FFS-Subgraph for TPSN, (c) only use temporal subgraphs obtained from uniform sampling. Table 4 shows the ablation experiment results of the "new-new" link prediction task in the Reddit dataset. First, only using the FFS-Subgraph for TPSN improves the performance from 91.28% to 95.09%. Only using the TFS-Subgraph for TPSN improves the performance from 91.28% to 96.35%. In particular, only using the TFS-Subgraph for TPSN is better

Table 4: Variants of Temporal Pattern Subgraph

| No. | Variant of Temporal Subgraph | link prediction |
|-----|------------------------------|-----------------|
| 1. | Subgraph of uniform sampling | 91.28 |
| 2. | FFS-Subgraph only | 95.09 |
| 3. | TFS-Subgraph only | 96.35 |
| 4. | **Ours** | **98.03** |

Table 5: Variants of Multi-subgraph Contrastive Learning

| No. | Variants of Multi-subgraph Contrastive Learning | node classification |
|-----|-------------------------------------------------|---------------------|
| 1. | **Ours**($N = 1$) | **69.21** |
| 2. | final embedding only ($N = 1$) | 66.32 |
| 3. | final embedding only ($N = 2$) | 67.81 |
| 4. | final embedding only ($N = 3$) | 69.77 |
| 5. | final embedding only ($N = 4$) | 69.79 |

than only using TFS-Subgraph in the model. This indicates the interactions from the Reddit dataset are influenced by the neighbors of the recent interactions more than those from the frequently interacting neighbors. Our model use both the TFS-Subgraph and the FFS-Subgraph gives a remarkable performance improvement of 7.75% comparing to the baseline. This stresses that capturing both types of temporal pattern is effective and important for temporal networks.

#### 4.3.2 VARIANTS OF MULTI-SUBGRAPH CONTRASTIVE LEARNING.

Here, we investigate the performance of different loss functions for contrastive learning. We conducted ablation studies on two variants, (a) the number of negative samples for each training node pair, and (b) whether to use the multi-subgraph embedding. Table 5 shows the ablation experiment results for node classification in the Reddit dataset. We only use the triplet loss for the final embedding in training, exceeding the AUC, our model needs $N = 3$, and is close to saturation at $N = 4$. This highlights that multi-subgraph contrastive learning is effective in reducing the time complexity while giving the same accuracy.

## 5 CONCLUSION

In this paper, our aim was to leverage the evolutionary pattern of nodes applied to inductive link prediction. We propose a novel framework, namely TPSN, for deep mining inductive temporal pattern representation learning, which has been overlooked in previous work. We adopt two types of subgraph to represent temporal patterns for nodes: a) the interactions which are closest to the current moment in the subgraph; and b) the interactions come from neighbors who interact frequently. In particular, we design the reconstruction adjacency algorithm, which positively improves the learning efficiency on temporal triadic closure law and reduces the possibility of oversmoothing. Furthermore, we use multi-subgraph contrastive learning to achieve higher accuracy with fewer negative samples. After experimental validation, the complete framework is effective in capturing temporal pattern signals in terms of the nodes and links. Also, ablation studies verify the effectiveness of each module proposed in the model. Finally, future work will focus on mining more temporally diverse patterns and identifying the optimal subgraph patterns for each real-world dataset.

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

## A    APPENDIX

---

**Algorithm 1:** Temporal Neighbors Finding Algorithm

---

**Input:** The set of nodes that need to query neighbors for current batch, $N$;
The set of deadlines that need to query neighbors for current batch, $T$;
**Output:** The set of neighbors for current batch, $Neighbor(N)$;

1  initial $Neighbor(N)$ ;
2  **for** $j = 1; j \leq len(C)$ **do**
3       $C$ = all edge of node N[j] adjacency list ;
4       **while** $left + 1 < right$ **do**
5           $mid = (left + right)//2$ ;
6           $curr_t = C[mid]$ ;
7           **if** $curr_t < T[j]$ **then**
8               $left = mid$;
9           **else**
10              $right = mid$;
11      **if** $T[right] < T[j]$ **then**
12          $Neighbor(N[j], T[j]) = C[: right]$;
13      **else**
14          $Neighbor(N[j]) = C[: left]$;
15 return $Neighbor(N)$ ;

---

---

**Algorithm 2:** Time First Search Subgraph Algorithm

---

**Input:** The set of neighbors for current batch, $Neighbor(N)$;
The set of nodes that need to query neighbors for current batch, $N$;
The max number of hop for nerighbor query, $K$;
The max number of subgraph size, $L$;
**Output:** The set of tfs-subgraph for current batch, $TFS - Subgraph(N)$
1   initial $TFS - Subgraph(N)$ ;
2   **for** *node j in N* **do**
3     one-hop adjacency list of node $j = Neighbor(j)$ ;
4     find k-hop Adjacency list of node $j$ by Algorithm. 1.;
5     Sort k-hop Adjacency list of node $j$ in chronological order;
6     take top-L edges to represent node $j$ 's TFS-Subgraph;
7   return $TFS - Subgraph(N)$ ;

---

**Algorithm 3:** Frequency First Search Subgraph Algorithm

---

**Input:** The set of neighbors for current batch, $Neighbor(N)$;
The set of nodes that need to query neighbors for current batch, $N$;
The max number of hop for nerighbor query, $K$;
The max number of subgraph size, $L$;
**Output:** The set of TFS-Subgraph for current batch, $FFS - Subgraph(N)$
1   initial $FFS - Subgraph(N)$ ;
2   **for** *node j in N* **do**
3     one-hop adjacency list of node $j = Neighbor(j)$ ;
4     find k-hop Adjacency list of node $j$ by Algorithm. 1.;
5     Sort k-hop Adjacency list of node $j$ in order of neighbor's frequency;
6     take top-L edges to represent node $j$ 's FFS-Subgraph;
7   return $FFS - Subgraph(N)$ ;

---

Table 6: Statistics of the datasets used in the experiments.

|  | Wikipedia | Reddit |
|---|---|---|
| # Nodes | 9,227 | 11,000 |
| # Edges | 157,474 | 672,447 |
| # Edge features | 172 | 172 |
| # Edge features type | LIWC | LIWC |
| Timespan | 30 days | 30 days |
| Chronological Split | 70%-15%-15% | 70%-15%-15% |
| # Nodes with dynamic labels | 216 | 366 |

