# OpenReview forum: "Inductive Representation Learning of Temporal Pattern Subgraphs in Temporal Networks"
_ICLR.cc/2026/Conference — Submitted to ICLR 2026_

### Official Review · Reviewer_vTit · 2025-10-23

**Soundness:** 2
**Presentation:** 1
**Contribution:** 2
**Rating:** 2
**Confidence:** 5

**Summary:**

This paper introduces TPSN, a novel framework for inductive representation learning in temporal graphs, designed to model dynamic relationships where test-time nodes may not have been observed during training. The method integrates information from two temporal pattern subgraphs, TFS-subgraph (capturing recent interactions) and FFS-subgraph (capturing frequent interactions), to derive node embeddings that effectively encode the laws of node evolution, such as the triadic closure principle.

To unify representations across multiple temporal subgraphs, TPSN employs a multi-subgraph contrastive learning strategy that aligns embeddings of nodes likely to interact in the future while maintaining separation between unrelated ones.

Empirical results across three downstream tasks, including transductive link prediction, inductive link prediction, and dynamic node classification, on two benchmark datasets (Reddit and Wikipedia) show that TPSN consistently outperforms baseline methods. Complementary ablation studies further validate the contribution of each component, highlighting the effectiveness of combining TFS and FFS subgraphs with contrastive learning for temporal pattern modelling.

**Strengths:**

- The paper introduces a novel inductive representation learning framework that effectively generalizes to unseen nodes in dynamic graphs by combining TFS-Subgraph (capturing recency-based temporal patterns) and FFS-Subgraph (capturing frequency-based temporal patterns) to model key laws of node evolution, such as triadic closure.

- The papers presents a new contrastive optimization approach across multiple temporal subgraph types, enabling TPSN to reconcile and unify embeddings from heterogeneous temporal patterns through multi-subgraph contrastive learning.

- On two benchmark datasets, TPSN consistently outperforms several state-of-the-art baselines across diverse evaluation settings, including transductive and inductive link prediction as well as dynamic node classification.

- Comprehensive ablation and attention analyses demonstrate the effectiveness and interpretability of the proposed components, showing that integrating multiple temporal subgraphs and the contrastive learning module significantly contributes to performance gains.

**Weaknesses:**

**W1: Wring and Readability.** The current manual script has several writing and formatting issues that hinder readability and comprehension.
- Please insert spaces between model names and author names, and use \citep to enclose author names in parentheses, clearly separating method names from citations.

-  Line 093: avoid repeating author names; use \cite instead of repeating to maintain clarity.

- The paper lacks a clear problem statement section. It is recommended to explicitly define the problem, the tasks addressed, and relevant definitions, especially for readers new to temporal graph learning, immediately before Section 3. This should include the definition of temporal graphs (currently Lines 139-143) and formal problem formulations for link prediction and node classification.

- Several variables and notations are used without prior definition, resulting in confusion. For example, in Algorithm 1, the variable “C” is used before being defined (Lines 2 and 3). Similarly, the index “i” appears undefined in Equations 1-3.

-  Notation throughout the paper is inconsistent. For instance, Line 215 uses “i” to denote a node, whereas Lines 195-198 use “u” and “v.” In Algorithm 1, “j” is an index, but in Algorithm 2, “j” refers to nodes. This inconsistency should be resolved for clarity.

- In Algorithm 1, the variable name “Neighbor(N)” is confusing, as it syntactically resembles a function call on N rather than a variable or data structure

**W2: Out-of-date Baselines.** The work compares the performance of TGT against out-of-date baselines. A naive but effective baseline, such as EdgeBank[1], and recent TGNN models[2,3,4]

**W3: Reproducibility.** Neither the source code for implementation nor the details about the hyperparameters (e.g learning rate, number of epochs, $\alpha$ used in Equation 5, etc.) used in experiments. These questions about the reproducibility of the work.

**W4: Dataset Diversity.** TPSN is evaluated only on two relatively limited datasets. For broader validation, consider including additional, larger-scale, and diverse benchmark datasets from the Temporal Graph Benchmark (TGB) [5] , covering more comprehensive link prediction and node property prediction scenarios.

**W5: Appendix Structure.** The appendix section would benefit from improved structure and should include detailed information about the baselines used, including their configurations and any implementation details relevant to fair comparison.


**Minor**
- The paper mainly reports AUC and AP for link prediction. Including Mean Reciprocal Rank (MRR) under the Temporal Graph Benchmark (TGB) evaluation settings would provide a more comprehensive view of TPSN 's ranking performance and enhance comparability with other temporal graph models evaluated on TGB[5].
---

[1] Poursafaei, Farimah, et al. "Towards better evaluation for dynamic link prediction." *Advances in Neural Information Processing Systems* 35 (2022): 32928-32941.

[2] Yu, Le, et al. "Towards better dynamic graph learning: New architecture and unified library." *Advances in Neural Information Processing Systems* 36 (2023): 67686-67700.

[3] Lu, Xiaodong, et al. "Improving temporal link prediction via temporal walk matrix projection." *Advances in Neural Information Processing Systems* 37 (2024): 141153-141182.

[4] Ding, Zifeng, et al. "Dygmamba: Efficiently modelling long-term temporal dependency on continuous-time dynamic graphs with state space models." *arXiv preprint arXiv:2408.04713* (2024).

[5] Huang, Shenyang, et al. "Temporal graph benchmark for machine learning on temporal graphs." Advances in Neural Information Processing Systems 36 (2023): 2056-2073.

**Questions:**

- It is unclear whether Algorithm 1 assumes that $C$ is chronologically sorted. If the correctness or stability of the algorithm depends on the temporal ordering of $C$, this requirement should be explicitly stated at the beginning.

- In Algorithm 2, please clarify the criterion for selecting the top‑k elements; is it based on time order, interaction frequency, or another factor?

- Lines 185–188: The explanation of how the law of triadic closure is captured through two different paths in the proposed graph matching algorithm is unclear. Can the authors elaborate on the mechanism, specifically, how these paths operationally encode or approximate triadic closure within the temporal subgraph structure?

- Line 194: If multiple paths exist between two nodes, how is the aggregated path distance computed, by the shortest, average, or total path length?

- Lines 195–198: It is unclear why node $v_i$  is replaced by node $A$ to create a new virtual edge. How does this help to reduce the possibility of oversmoothing?

- Figure 3: The meaning of “layer” is unclear. Does it refer to neighbor hops or network layers? Please clarify the terminology and ensure it is used consistently throughout the paper.

- Line 266: The term “negative sample” is unclear. Does it refer to a negative edge or another sampling strategy?

- Equation 6: The model’s main training task is unclear. Is it trained with positive and negative edges? Are nodes $i$
 and $j$ positive or negative pairs? Also, $h_n$ is undefined; can the author clarify this?


- Section 4.1.4: The values for the number of 1-hop neighbors $N$ nd subgraph size $M$ are not specified. Please clarify their values and discuss how varying$N$ and $M$ affects model performance and computational cost.

- Could the authors provide a detailed runtime analysis of TPSN compared to baseline methods? Specifically, how do the individual components, including TFS-subgraph generation, FFS-subgraph generation, and adjacent edge reconstruction, scale with increasing graph size, especially on large-scale datasets like those in TGB[5]?
- It is not clear how the model is evaluation, is it evaluated with positive and negative egdes?

- Tables 4 and 5: Are the results from a single run with one random seed? Please provide standard deviations to understand the variability and robustness of the results.

- In Table 5, why is the result for$N=1$ bolded when $N=3$ and $N=4$ show better performance? This appears to contradict the claim on line 269 that the model achieves better results without increasing the number of negative samples. Could the authors please clarify?

- DyGFormer[2] also considers recency and frequency of neighbors through a neighbor co-occurrence encoding scheme and patching technique. Could the authors clarify what differentiates TPSN from DyGFormer, particularly regarding how TPSN’s temporal pattern subgraph mining and multi-subgraph contrastive learning offer advantages over DyGFormer’s sequence-based Transformer approach?


---

[2] Yu, Le, et al. "Towards better dynamic graph learning: New architecture and unified library." *Advances in Neural Information Processing Systems* 36 (2023): 67686-67700.

[5] Huang, Shenyang, et al. "Temporal graph benchmark for machine learning on temporal graphs." Advances in Neural Information Processing Systems 36 (2023): 2056-2073.

---

### Official Review · Reviewer_WyES · 2025-10-25

**Soundness:** 2
**Presentation:** 1
**Contribution:** 2
**Rating:** 2
**Confidence:** 4

**Summary:**

This paper proposes a framework for inductive representation learning on temporal networks named TPSN. The approach extracts two types of temporal pattern subgraphs: TFS-Subgraph, which uses Time First Search to capture recent interactions, and FFS-Subgraph, which uses Frequency First Search to capture high-frequency interactions. The method performs adjacent edge reconstruction based on triadic closure laws and employs multi-subgraph contrastive learning. Experiments on Reddit and Wikipedia datasets demonstrate improvements over baselines in link prediction and node classification tasks.

**Strengths:**

- The paper addresses an important problem to learn representations in temporal graphs by capturing two distinct temporal patterns that reflect realistic social network dynamics.
- The proposed method based on the temporal patterns is technically sound.
- The experiments are conducted across multiple tasks (transductive/inductive link prediction, node classification) with detailed ablation studies validating the effectiveness of the proposed method.

**Weaknesses:**

- The evaluation is restricted to only two datasets (Reddit and Wikipedia), which leads to two concerns:
  1. The graphs are not diverse enough. Testing on diverse temporal networks (e.g., financial, biological, transportation) would strengthen generalizability claims.
  2. These two datasets are relatively small, 9-10k nodes, which cannot fully demonstrate the efficiency of the proposed method.
- The paper lacks comparisons with more recent temporal GNN methods and subgraph-based approaches. The baselines are somewhat dated (most from 2018-2020), and missing comparisons with more recent methods such as DyGFormer [1] and GraphMixer [2].

- While the approach is intuitive, the paper lacks theoretical analysis of why this particular combination of TFS and FFS patterns is optimal, how the reconstruction affects information flow mathematically, or convergence guarantees for the contrastive learning objective.

- There is no complete complexity analysis of the proposed method (only the FFS-subgraph detection component contains the complexity of subgraph pattern mining).

- The paper contains grammatical errors and unclear explanations.
  1. The format of reference: Throughout the paper, citations lack proper punctuation between consecutive references, and the reference after text requires a bracket.
  2. Some explanations are not clear. For example, the motivation for why edge reconstruction specifically helps with triadic closure learning is not clearly discussed. The results of attention analysis is not very clear.

[1] Yu L, Sun L, Du B, et al. Towards better dynamic graph learning: New architecture and unified library[J]. Advances in Neural Information Processing Systems, 2023, 36: 67686-67700.

[2] Cong W, Zhang S, Kang J, et al. DO WE REALLY NEED COMPLICATED MODEL ARCHITECTURES FOR TEMPORAL NETWORKS?[C]//11th International Conference on Learning Representations, ICLR 2023. 2023.

**Questions:**

1. Complexity of the proposed method
2. Experimental results on more diverse and larger-scale graphs
3. Compareison with more recent baselines
4. More clear explanations of model design and experimental results

---

### Official Review · Reviewer_DLTe · 2025-10-29

**Soundness:** 2
**Presentation:** 2
**Contribution:** 2
**Rating:** 2
**Confidence:** 4

**Summary:**

This paper proposes the TPSN framework for inductive representation learning on temporal graphs, aiming to capture temporal subgraph patterns through time-first and frequency-first mining, adjacency reconstruction, and multi-subgraph contrastive learning. The method is evaluated on Reddit and Wikipedia datasets for link prediction and node classification tasks.

**Strengths:**

- The paper addresses a relevant problem in temporal graph representation learning and provides a structured pipeline for subgraph-based pattern extraction.

- The framework integrates multiple components—subgraph mining, positional encoding, and contrastive learning—into a unified design.

- The experimental section includes comparisons across several benchmark datasets and ablation studies for individual modules

**Weaknesses:**

- The claimed methodological innovation is marginal, with many components (e.g., attention-based aggregation, triplet loss) borrowed from prior work such as TGAT and DynamicTriad without substantial conceptual advancement.
﻿
- Experimental settings are limited to small-scale datasets, leaving scalability, robustness, and generalizability untested.
The TFS and FFS subgraph detection algorithms appear to involve recursive neighborhood expansion and edge sorting. What is the actual runtime and memory overhead compared to conventional temporal motif mining? Are these steps parallelizable or optimized in implementation?

**Questions:**

- The TFS and FFS subgraph detection algorithms appear to involve recursive neighborhood expansion and edge sorting. What is the actual runtime and memory overhead compared to conventional temporal motif mining? Are these steps parallelizable or optimized in implementation?
﻿
- The adjacency reconstruction process seems to generate additional virtual edges. How does this affect the memory footprint and computational cost of the model, particularly in dense temporal networks?
﻿
- The framework incorporates multi-subgraph contrastive learning with triplet loss. Could the authors quantify the additional training cost (e.g., time per epoch or GPU memory) compared with TGAT or JODIE under the same hardware setup?
﻿
﻿
- The claim that TPSN achieves better performance with fewer negative samples is interesting. Could the authors explain the trade-off between accuracy and time complexity when varying the number of negative samples?

---

### Official Review · Reviewer_zZve · 2025-11-01

**Soundness:** 2
**Presentation:** 2
**Contribution:** 2
**Rating:** 2
**Confidence:** 3

**Summary:**

The paper introduces TPSN, a novel GNN layer for temporal graphs, which is designed to explicitly encoding higher-order structures using an efficient subgraph-mining algorithm.
The main part of the work (Section 3) is devoted to the introduction of this encoding, which is then combined with a contrastive learning mechanism (Section 3.4).
Experimental validation confirms the effectiveness of TPSN.

**Strengths:**

- The modelling of high-order interactions in temporal graphs is a highly relevant topic.
- The experiments are convincingly demonstrating the promising results of the method.

**Weaknesses:**

- Section 3 is the central part, introducing the new architecture. It is however a quite lenghty text descripion, without a sufficiently clear formalization. This makes it hard to reproduce the work, and to understand its novelty and impact.

- Part of the work is motivated by ideas typical of social interactions and complex systems. These claims are however not substantiated by references to the literature (see e.g. the second paragraph in the introduction), and no efforts are made to (at least partially) survey the existing work in this direction (see e.g. [5, 6]).

- The related work and the considered baselines are not up-to-date (the newest architecture are DynAERNN and TGAT). In particular, the work almost discards an entire family of event-based architectures (TGL [1], APAN [2], DGNN [3], TGN [4], to mention a few). These choices severly limit the scope of the work, and make the experimental results not sufficiently convincing.


[1] H. Zhou et al., TGL: A general framework for temporal GNN training on billion-scale graphs, Proc. VLDB Endow. (2022).

[2]  X. Wang et al., APAN: Asynchronous propagation attention network for real-time temporal graph embedding, SIGMOD (2021).

[3] Y. Ma et al., Streaming graph neural networks, SIGIR (2020).

[4] E. Rossi et al., Temporal graph networks for deep learning on dynamic graphs (2020)

[5] Liu, J., et al., Higher-order Structure Boosts Link Prediction on Temporal Graphs, arXiv (2025).

[6] C. Battiloro et al., Generalized simplicial attention neural networks, IEEE TSIPN (2024).

**Questions:**

Apart from the points discussed above, there are several typographic errors (already in the abstract).

---

### Meta-Review · Area_Chair_5Pzw · 2026-01-05

**Summary:**

Reviewers are all negative about this work due to some major issues.

**Reviewer Scores:**

n/a

---

### Decision · Program_Chairs · 2026-01-26

Reject